

# Early skin-to-skin contact between healthy late preterm infants and their parents: an observational cohort study

Kerstin H. Nyqvist[1,*], Andreas Rosenblad[2,*], Helena Volgsten[3,*], Eva-Lotta Funkquist[1] and Elisabet Mattsson[1,4]

[1] Department of Women's and Children's Health, Uppsala University, Uppsala, Sweden
[2] Center for Clinical Research Västerås, Uppsala University, Västerås, Sweden
[3] Department of Public Health and Caring Sciences, Uppsala University, Uppsala, Sweden
[4] Department of Health Care Sciences, Ersta Sköndal Bräcke University College, Stockholm, Sweden
[*] These authors contributed equally to this work.

Corresponding author
Elisabet Mattsson,
elisabet.mattsson@kbh.uu.se

## ABSTRACT

**Background**. Skin-to-skin contact (SSC) is an important factor to consider in the care of late preterm infants (born between 34 0/7 and 36 6/7 completed weeks of gestation). The literature suggests that SSC between preterm infants and their mothers facilitates breastfeeding. However, more studies are needed to explore potential dose-response effects between SSC and breastfeeding as well as studies that explicitly investigate SSC by fathers among late preterm infants. The aim was to investigate the duration of healthy late preterm infants' SSC with the mother and father, respectively, during the first 48 h after birth and the associations with breastfeeding (exclusive/partial at discharged), clinical and demographic variables.

**Methods**. This was an observational cohort study in which parents to healthy late preterm infants, born between 34 5/7 and 36 6/7 completed weeks of gestation, recorded duration of SSC provided by mother and father, respectively. Demographic and clinical variables were retrieved from the medical records and were used as predictors. Multiple linear regression analysis was used to assess the association between the predictors and the outcome, SSC (hours), separately for mothers and fathers.

**Results**. The mean (standard deviation [SD]) time per day spent with SSC with mothers ($n = 64$) and fathers ($n = 64$), was 14.7 (5.6) and 4.4 (3.3) hours during the first day (24 h) after birth and 9.2 (7.1) and 3.1 (3.3) hours during the second day (24 h), respectively. Regarding SSC with mothers, no variable was significantly associated with SSC during the first day, while the mean (95% confidence interval [CI]) time of SSC during the second day was 6.9 (1.4–12.4) hours shorter for each additional kg of birthweight ($p = 0.014$). Concerning SSC with fathers, the mean (95% CI) time of SSC during the first day was 2.1 (0.4–3.7) hours longer for infants born at night ($p = 0.015$), 1.7 (0.1–3.2) hours longer for boys ($p = 0.033$), 3.2 (1.2–5.2) hours longer for infants born by caesarean section ($p = 0.003$), and 1.6 (0.1–3.1) hours longer for infants exclusively breastfed at discharge ($p = 0.040$). During the second day, the mean (95% CI) time of SSC with fathers was 3.0 (0.6–5.4) hours shorter for each additional kg of birthweight ($p = 0.014$), 2.0 (0.5–3.6) hours longer for infants born during night-time ($p = 0.011$), 2.9 (1.4–4.4) hours longer if the mother was primipara ($p < 0.001$), and 1.9 (0.3–3.5) hours shorter if supplementary artificial milk feeds were given. None of the other predictors, i.e., mother's age, gestational age, or induction of labor were

significantly associated with infants' SSC with mothers or fathers during any of the first two days after birth.

**Conclusion**. Future studies are warranted that investigate duration of SSC between late preterm infants and their parents separately and the associations with breastfeeding and other variables of clinical importance.

## INTRODUCTION

Late preterm infants, defined as infants born between 34 0/7 and 36 6/7 completed weeks of gestation (*Committee on Obstetric Practice, 2008*), are to be considered physiologically immature and should be diligently evaluated, monitored, and followed by health care professionals who are educated to monitor and prevent vulnerabilities associated with prematurity (*Raju et al., 2006*). Even though late preterm infants are born only a few weeks early, they are at an increased risk for bradycardia, feeding difficulties, hypoglycemia, jaundice, respiratory distress, sepsis, temperature instability, and hospital readmission (*Engle et al., 2007*; *Celik et al., 2013*). Lower breastfeeding rates at discharge have been found in late preterms, compared with infants born at term (*Ayton et al., 2012*).

Skin-to-skin contact (SSC), the infant placed naked (with a diaper and with or without a cap) in skin-to-skin contact in a vertical position on the caregiver's bare chest (*WHO, 2013*), is an important factor to consider in the care of late preterm infants. It has been hypothesised that SSC between preterm infants and their mothers facilitates breastfeeding and breastmilk production, maintains infant body temperature, and enhances infant long-term cognitive development (*Britton et al., 2007*; *Nyqvist et al., 2010*; *Feldman, Rosenthal & Eidelman, 2014*). Furthermore, evidence supports the use of SSC to promote breastfeeding among healthy infants, and the literature indicates that women practicing SSC after cesarean section are more likely to breastfeed up to four months after birth (*Moore et al., 2016*). Given the health benefits that breast milk confers to preterm infants (*American Academy of Pediatrics, 2012*) there is a paucity of studies concerning breastfeeding outcomes related to SSC among late preterm infants (*Moore et al., 2016*).

It is important to distinguish between SSC for healthy infants and kangaroo mother care (KMC). Continuous or intermittent SSC is one of three components of KMC. The other two include exclusive or nearly exclusive breastfeeding and early discharge from hospital (*Cattaneo et al., 1998*; *WHO, 2013*). The World Health Organization (WHO) guidelines regarding preterm birth outcomes recommend KMC for thermal care for clinically stable newborns weighing 2,000 g or less at birth (*WHO, 2015*). However, KMC is seldom practiced in its entirety. Consequently, studies regarding KMC often focus on SSC as the key intervention (*Conde-Agudelo & Días-Rossello, 2016*).

So far, research has mainly focused on psychological and physiological effects related to early mother-infant SSC (*Conde-Agudelo & Días-Rossello, 2016*; *Moore et al., 2016*).

However, a recent study concludes that no differences regarding physiological and stress responses between SSC provided by fathers versus mothers could be identified among preterm infants (*Srinath et al., 2016*). Furthermore, twelve studies investigating SSC by fathers and its impact on infant and paternal outcomes have recently been reviewed (*Shorey, He & Morelius, 2016*). The authors conclude that father-infant SSC had a positive impact on infants' behavioral response, bio-physiological markers, pain and temperature regulation, as well as paternal distress, interaction behaviour and role attainment.

To summarise, there is a need for further studies that explicitly investigate SSC by fathers (*Shorey, He & Morelius, 2016*). Additionally, the literature regarding SSC highlights that more studies are needed to explore possible dose-response effects between SSC and breastfeeding outcomes (*Moore et al., 2016*). Against this background, the aim of the present study was to investigate the duration of healthy late preterm infants' SSC with the mother and father, respectively, during the first 48 h after birth and its associations with breastfeeding, clinical and demographic variables.

## METHODS

### Study design

This was an observational cohort study in which parents to late preterm infants recorded duration of SSC provided by mother and father, respectively.

Late preterm infants born between 35–36 completed weeks of gestation were screened for inclusion. Eligibility criteria included: Swedish-speaking parents of a healthy late preterm infant cared for in family rooms at the postnatal ward of Uppsala University Hospital in Uppsala, Sweden. For the purpose of this study, "healthy" means that the infant spent no time in the neonatal intensive care unit, i.e., had no respiratory morbidity, neonatal infection or feeding difficulties that required nasogastric feeds. Neither did the infant require active management for hypothermia (incubator) or support with intravenous fluids due to hypoglycaemia. Exclusion criteria included: multiple births, infants whose mother chose not to breastfeed and infants who needed neonatal intensive care.

### Study context

The proportion of infants born prematurely has been relatively constant during the past decades in Sweden, that is, nearly 5 percent of all single births. Approximately 4 percent are born between 33–36 completed weeks, whereas one percent is born between 22–32 completed weeks of gestation (*Socialstyrelsen, 2017*). Approximately 230 late preterm infants are born each year at Uppsala University Hospital in Uppsala, Sweden.

At the time of the study, healthy late preterm infants born at ≥35 completed weeks of gestation were routinely cared for at the postnatal wards, i.e., together with term infants. Late preterm infants born <35 completed weeks of gestation were routinely cared for at the neonatal intensive care unit, and thus were excluded from the present study. Neonatologists directed the infants' medical care plan at the postnatal wards, whereas a midwife working with a nurse assistant (a person who has completed a brief health care training program and provides support services for registered nurses/midwives) provided the midwifery care. The midwife was responsible for providing care for the infant and the mother,
including the detection of complications, the accessing of medical care, breastfeeding advice and support, as well as postnatal health counselling and education. The care was family centered, i.e., the majority of fathers/partners stayed together with the mother and the new-born baby in family rooms during the postnatal care period. Siblings had free visiting hours, but were not allowed to stay overnight. Mothers intending to breastfeed were offered individualised support from midwives to facilitate the establishment of lactation and exclusive breastfeeding. According to standard care, the infants were placed SSC on their mother immediately after birth to enable the occurrence of the first breastfeed as soon as possible. If the mother was restricted due to medical reasons from holding the baby after birth, the infant was placed SSC with the father/partner. Health professionals recommended SSC between the infant and the mother/father, i.e., to hold the baby naked (with a diaper and with or without a cap) in the kangaroo position (*WHO, 2013*) on the caregiver's bare chest during the hospital stay. To manage safe co-sleeping at night, information to parents was based on guidelines from the Swedish National Board of Health and Welfare. The guidelines included placing the infant in a supine position, making sure that the baby had enough space to move, ensuring that the baby did not get too warm and that the infant's head was not covered. At the time of the study, healthy late preterm infants stayed, on average, for four days at the postnatal ward (*Mattsson et al., 2015*).

## Study sample

Ethical approval to conduct the study was obtained from the Regional Ethical Review Board in Uppsala, Sweden (approval number 2010/287).

See Fig. 1 for numbers of potential participants, participants and reasons for exclusion and attrition.

During the inclusion period, between the 15th of November 2010 and the 30th of November 2011, parents of 86 late preterm infants agreed to participate in the study. However, five infants were excluded as they were referred to the neonatal intensive care unit and parents of 17 infants did not complete the data collection. Thus, parents of 64 (61%) late preterm infants completed the data collection. Forty-three infants were born between 36 0/7 and 36 6/7 completed weeks of gestation, 20 infants between 35 0/7 and 35 6/7 and one infant was born at 34 5/7 completed weeks. Due to the real life design of the study, no power calculation was performed.

## Data collection

Potential participants i.e., late preterm infants cared for in family rooms at the postnatal wards, were identified through the log book at the hospital's maternity ward. The book includes name of the ward where the mother/infant were cared for after birth, the mother's name, Swedish identification number, address, date of birth and completed weeks of gestation. Written and oral information about the study was given to parents by the authors EM, ELF, and HV within 24 h after birth at the postnatal ward. Parents who agreed to participate and completed a written informed consent form began completing the diary, one chart per day, commencing from the time they gave their consent, during the infant's stay at hospital. Data from the time of birth to the initiation of study participation were

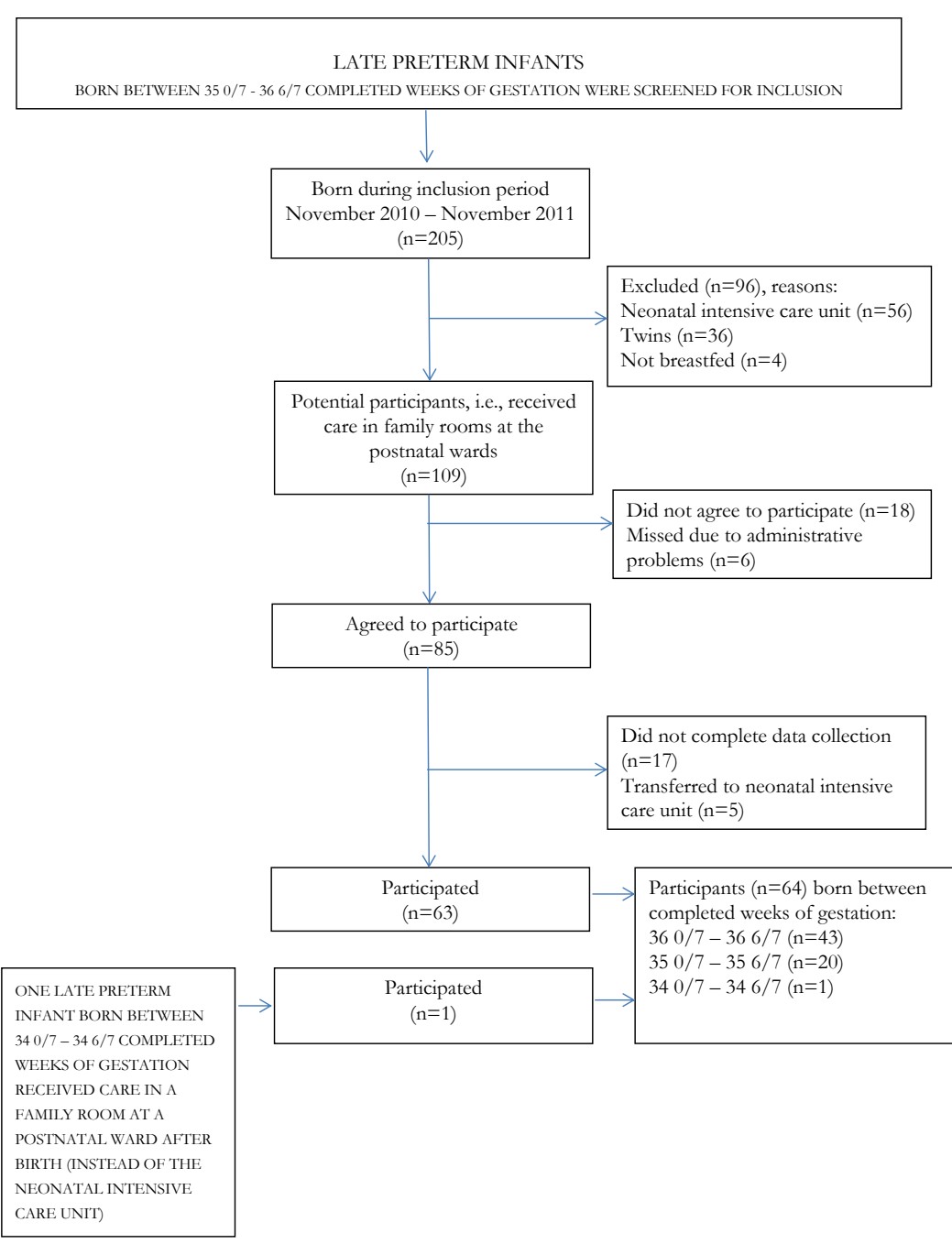

**Figure 1   Numbers of potential participants, participants, reasons for exclusion and attrition.**

entered retrospectively in the diary by the parents. When the family was discharged from the hospital, the diary was left in a special study mailbox at the postnatal ward. One of the authors (EM) retrieved the study variables from the medical records.

Data were collected via a parental diary (*Blomqvist, Rubertsson & Nyqvist, 2011*; *Mattsson et al., 2015*). The diary consisted of one chart for each day with a timeline (hours and minutes) and columns with fixed options regarding the timing and duration of SSC, breastfeeding and supplementary artificial milk feeds. Parents recorded the duration of each breastfeeding session, supplementary artificial milk feeds and SSC provided by mother/father/others by drawing a line marking the hours and minutes for each variable. The demographic characteristics mother's age (years) and infant's sex (boy/girl), and the clinical variables gestational age (completed weeks and days), infant's birthweight (kg), time of birth (day-time: 07:00–18:59/night-time: 19:00–06:59), parity (primipara/multipara), induction of childbirth (yes/no), mode of delivery (vaginal delivery/caesarean section), supplemental feeding (given/not given), and breastfeeding at discharge (partial/exclusive), were used as predictors. Other clinical variables of importance, such as signs of fetal asphyxia according to fetal heart rate monitoring ($n = 2$), vacuum extraction ($n = 2$), manual extraction of placenta ($n = 1$), large perineal tears ($n = 1$), and postpartum bleeding >1,000 ml ($n = 1$) were assessed but were not used in the analyses due to too few observations to perform any relevant statistical analyses. Demographic and clinical variables were chosen on the basis of the available literature at the time of the study (*Britton et al., 2007*; *Erlandsson et al., 2007*; *Raju et al., 2006*; *Walker, 2008*) and clinical experiences of the members in the research group. Breastfeeding at discharge was defined according to the World Health Organization (WHO) definition of exclusive breastfeeding; that is, the infant received only breast milk without any additional food or drink (not even water) at discharge.

Data regarding SSC during the first 48 h after birth are presented in the present study, whereas data regarding supplementary artificial milk feeds related to breastfeeding and associated clinical parameters have been presented elsewhere (*Mattsson et al., 2015*).

## Statistical analyses

The outcome of interest in the present study was the duration of the infant's SSC with the mother and father during the first and second days (24 h), respectively, after birth and the associations with breastfeeding, clinical and demographic variables.

Sample characteristics are described by frequencies and percentages, $n$ (%), for categorical variables and by means and standard deviations (SD), supplemented by median and range, for continuous variables. Linear regression analyses were used for assessing the associations between the predictors and the outcome, separately for mothers and fathers. Three kinds of regression models were used: (i.) the unadjusted model, using each predictor in separate regression models; (ii.) the full model, including all predictors simultaneously in the same regression model; and (iii.) the full model, using a backward elimination approach starting with the full model and step-by-step deletion of the variable with the highest $p$-value, re-estimating the model and deleting the variable which now had the highest $p$-value, and so on, with the process continuing until only variables with

*p*-values <0.20 remained in the model (*Newbold, 1995*). No adjustments were made for multiple comparisons. IBM SPSS Statistics 22/23/24 (IBM Corp., Armonk, NY) was used for statistical analyses. Two-sided *p*-values <0.05 were considered statistically significant.

## RESULTS

Parents' of 64 healthy late preterm infants completed the data collection about SSC. Characteristics of the study sample and the care provided are given in Table 1.

The mothers' median (range) age was 30.5 (19–40) years. Thirteen mothers (20.3%) gave birth by caesarean section, of which two (3.1%) were elective. The median (range) of infants' gestational age was 36 2/7 (34 5/7–36 6/7) weeks. No infant had an Apgar score below 7 points at 5 min after birth.

During the first day after birth, the infants' median time spent SSC with mothers ($n = 64$) was 14.8 h (range 0.0–24.0) and 4.3 h (range 0.0–14.0) with fathers. The second day the corresponding figures were 9.0 h (range 0.0–24.0) SSC with mothers and 2.5 h (range 0.0–13.0) SSC with fathers.

The associations between the demographic and clinical variables, on the one hand, and the infants SSC with mothers and fathers during the first and second 24 h after birth on the other hand, are presented in Tables 2 and 3, respectively.

None of the demographic or clinical variables were significantly associated with SSC with mothers during the first day. However, the infant's birthweight was strongly associated with SSC with mothers during the second day, with a mean (95% confidence interval [CI]) of 6.9 (1.4–12.4) hours shorter SSC for each additional kg of birthweight for the trimmed model ($p = 0.014$), i.e., infants with lower birthweights had more SSC than infants with higher birthweights (Table 2).

Among fathers, time of birth, infant's gender, mode of delivery, and exclusive breastfeeding at discharge were all significantly associated with SSC during the first day in the trimmed model, with a mean (95% CI) of 2.1 (0.4–3.7) hours longer SSC for infants born at night ($p = 0.015$), 1.7 (0.1–3.2) hours longer SSC if the infant was a boy ($p = 0.033$), 3.2 (1.2–5.2) hours longer SSC if the infant was born by caesarean section ($p = 0.003$), and 1.6 (0.1–3.1) hours longer SSC if the infant was exclusively breastfed at discharge ($p = 0.040$). During the second day, the infant's birthweight, time point of birth, parity, and if supplemental feeding were given, were all significantly associated with SSC with the father in the trimmed model, with a mean (95% CI) of 3.0 (0.6–5.4) hours shorter SSC for each additional kg of birthweight ($p = 0.014$), 2.0 (0.5–3.6) hours longer SSC for infants born at night ($p = 0.011$), 2.9 (1.4–4.4) hours longer SSC if the mother was primipara ($p < 0.001$), and 1.9 (0.3–3.5) hours shorter SSC for infants given supplementary artificial milk feeds ($p = 0.024$) (Table 3).

None of the following variables were associated with infants' SSC with mothers (Table 2) or fathers (Table 3) during any of the first two days after birth: mother's age, gestational age, or induction of childbirth.

**Table 1** Duration of SSC with mothers ($n = 64$) and fathers ($n = 64$) and description of demographic and clinical variables.

| Variables | Mean ± SD/Proportions |
|---|---|
| *Skin-to-skin contact* | |
| Duration the first day after birth (24 h) | |
|   - Mothers (h) | 14.7 ± 5.6 |
|   - Fathers (h) | 4.4 ± 3.3 |
| Duration the second day after birth (24 h) | |
|   - Mothers (h) | 9.2 ± 7.1 |
|   - Fathers (h) | 3.1 ± 3.3 |
| *Demographic variables* | |
| Mother's age (years) | 30.3 ± 5.0 |
| Infant's gender | |
|   - Girl | 48.4% |
|   - Boy | 51.6% |
| *Clinical variables* | |
| Parity | |
|   - Primipara | 45.3% |
|   - Multipara | 54.7% |
| Induction of childbirth | |
|   - Yes | 20.3% |
|   - No | 79.7% |
| Mode of delivery | |
|   - Vaginal delivery | 79.7% |
|   - Caesarean section | 20.3% |
| Time point of birth (24-hour clock) | |
|   - Day-time (07:00–18:59) | 53.1% |
|   - Night-time (19:00–06:59) | 46.9% |
| Gestational age (weeks) | 36.2 ± 0.6 |
|   - 34 5/7–35 6/7 | 32.8% |
|   - 36 0/7–36 6/7 | 67.2% |
| Infant's birthweight (kg) | 2.9 ± 0.3 |
| Supplementary artificial milk feeds | |
|   - Given | 65.6% |
|   - Not given | 34.4% |
| Breastfeeding at discharge | |
|   - Partial | 53.1% |
|   - Exclusive[a] | 46.9% |

**Notes.**

SD, standard deviation.

[a] The infant received only breast milk without any additional food or drink, not even water, at discharge.

**Table 2  Results from multiple linear regression for associations with infants' SSC with mothers the first and second day (24 h) after birth, respectively.**  Significant p-values are printed in bold.

| Day | Variable | Unadjusted β (95% CI) | P | Full model[a] β (95% CI) | P | Trimmed model[b] β (95% CI) | P |
|---|---|---|---|---|---|---|---|
| First | Mother's age (years) | 0.019 (−0.264: 0.302) | 0.893 | 0.036 (−0.299; 0.371) | 0.831 | −[c] | |
| | Boy | −2.617 (−5.352; 0.119) | 0.061 | −2.881 (−5.936; 0.174) | 0.064 | −2.658 (−5.413; 0.098) | 0.058 |
| | Primipara | −1.641 (−4.437; 1.154) | 0.245 | −1.146 (−4.286; 1.994) | 0.467 | −[c] | |
| | Induction of childbirth | 1.126 (−2.360; 4.611) | 0.521 | −0.394 (−4.143; 3.356) | 0.834 | −[c] | |
| | Caesarean section | −2.084 (−5.541; 1.373) | 0.233 | −3.250 (−7.322; 0.821) | 0.115 | −2.229 (−5.590; 1.133) | 0.190 |
| | Night-time birth | −0.030 (−2.849; 2.789) | 0.983 | −0.987 (−4.146; 2.173) | 0.534 | −[c] | |
| | Gestational age (weeks) | −0.396 (−2.928: 2.136) | 0.756 | −0.535 (−3.599; 2.530) | 0.728 | −[c] | |
| | Infant's birthweight (kg) | −0.692 (−5.136; 3.752) | 0.757 | −1.789 (−6.798; 3.221) | 0.477 | −[c] | |
| | Supplementary artificial milk feeds given | −1.549 (−4.485; 1.387) | 0.296 | −2.164 (−6.320; 1.992) | 0.301 | −2.624 (−6.123; 0.875) | 0.139 |
| | Exclusive breastfeeding at discharge | −1.191 (−3.994; 1.612) | 0.399 | −2.601 (−6.262; 1.060) | 0.160 | −3.186 (−6.458; 0.087) | 0.056 |
| Second | Mother's age (years) | −0.164 (−0.523; 0.196) | 0.366 | −0.245 (−0.662; 0.173) | 0.245 | −[c] | |
| | Boy | −3.823 (−7.286; −0.360) | **0.031** | −2.947 (−6.756; 0.862) | 0.127 | −3.276 (−6.757; 0.206) | 0.065 |
| | Primipara | −0.972 (−4.574; 2.630) | 0.592 | −1.573 (−5.488; 2.342) | 0.424 | −[c] | |
| | Induction of childbirth | 0.087 (−4.380; 4.554) | 0.969 | −2.884 (−7.559; 1.791) | 0.221 | −[c] | |
| | Caesarean section | −0.275 (−4.742; 4.192) | 0.902 | −0.850 (−5.926; 4.227) | 0.738 | −[c] | |
| | Night-time birth | −1.421 (−5.005; 2.162) | 0.431 | −1.356 (−5.295; 2.583) | 0.493 | −[c] | |
| | Gestational age (weeks) | −0.382 (−3.618; 2.853) | 0.814 | −1.222 (−5.042; 2.599) | 0.524 | −[c] | |
| | Infant's birthweight (kg) | −6.321 (−11.888; −0.755) | **0.027** | −8.482 (−14.727; −2.236) | **0.009** | −6.941 (−12.444; −1.438) | **0.014** |
| | Supplementary artificial milk feeds given | −2.425 (−6.159; 1.309) | 0.199 | −4.097 (−9.279; 1.085) | 0.119 | −2.530 (−6.262; 1.202) | 0.180 |
| | Exclusive breastfeeding at discharge | 0.555 (−3.044; 4.154) | 0.759 | 2.376 (−6.940; 2.189) | 0.301 | −[c] | |

**Notes.**
[a] First day: $R^2 = 0.171$; second day $R^2 = 0.241$.
[b] First day: $R^2 = 0.142$; second day $R^2 = 0.177$.
[c] Not in model.

## DISCUSSION

This is one of the first studies that explicitly investigate duration of SSC between healthy late preterm infants and their mothers and fathers separately and its associations with breastfeeding, clinical and demographic variables.

Breastfeeding at discharge was not associated with the extent of infants' SSC with mothers during any of the first two days after birth. However, all mothers in our study were breastfeeding to some extent at hospital discharge, 47% exclusively. Research regarding dose-response effect between duration of SSC and breastfeeding is ambiguous (*Bramson et al., 2010*; *Moore et al., 2016*). In contrast to our results, which present the dose-response association during a 24-hour day, *Bramson et al. (2010)* found a clear dose-response association between mother-infant duration of SSC for the first 3 h following birth and exclusive breastfeeding during hospitalisation. *Moore et al. (2016)*, on the other hand, demonstrated in their review no differences between less versus more than 60 min of SSC

Table 3 Results from multiple linear regression for associations with infants' SSC with fathers the first and second day (24 h) after birth, respectively. Significant *p*-values are printed in bold.

| Day | Variable | Unadjusted $\beta$ (95% CI) | P | Full model[a] $\beta$ (95% CI) | P | Trimmed model[b] $\beta$ (95% CI) | P |
|---|---|---|---|---|---|---|---|
| First | Mother's age (years) | 0.001 (−0.165; 0.167) | 0.992 | −0.050 (−0.230; 0.131) | 0.583 | –[c] | |
| | Boy | 1.512 (−0.096; 3.121) | 0.065 | 1.907 (0.262; 3.551) | **0.024** | 1.678 (0.137; 3.220) | **0.033** |
| | Primipara | 1.701 (0.097; 3.304) | **0.038** | 1.363 (−0.327; 3.053) | 0.112 | 1.297 (−0.316; 2.910) | 0.113 |
| | Induction of childbirth | 0.546 (−1.503; 2.596) | 0.596 | 1.214 (−0.804; 3.233) | 0.233 | 1.278 (−0.647; 3.203) | 0.189 |
| | Caesarean section | 1.753 (−0.253; 3.759) | 0.086 | 3.273 (1.081; 5.464) | **0.004** | 3.177 (1.153; 5.202) | **0.003** |
| | Night-time birth | 0.676 (−0.971; 2.323) | 0.415 | 2.036 (0.336; 3.737) | **0.020** | 2.051 (0.413; 3.689) | **0.015** |
| | Gestational age (weeks) | 0.169 (−1.319; 1.657) | 0.821 | 0.508 (−1.141; 2.158) | 0.539 | –[c] | |
| | Infant's birthweight (kg) | −2.672 (−5.255; −0.089) | **0.043** | −2.390 (−5.086; 0.306) | 0.081 | −1.956 (−4.451; 0.539) | 0.122 |
| | Supplementary artificial milk feeds given | 0.182 (−1.557; 1.922) | 0.835 | −0.419 (−2.656; 1.818) | 0.709 | –[c] | |
| | Exclusive breastfeeding at discharge | 0.896 (−0.745; 2.536) | 0.279 | 1.068 (−0.902; 3.039) | 0.282 | 1.594 (0.073; 3.114) | **0.040** |
| Second | Mother's age (years) | −0.108 (−0.274; 0.057) | 0.195 | −0.096 (−0.270; 0.077) | 0.270 | −0.108 (−0.265; 0.049) | 0.174 |
| | Boy | 0.047 (−1.619; 1.714) | 0.955 | 0.015 (−1.568; 1.597) | 0.985 | –[c] | |
| | Primipara | 2.884 (1.380; 4.388) | **<0.001** | 2.853 (1.227; 4.480) | **0.001** | 2.895 (1.380; 4.411) | **<0.001** |
| | Induction of childbirth | 0.934 (−1.123; 2.991) | 0.367 | 0.447 (−2.390) | 0.646 | –[c] | |
| | Caesarean section | −0.104 (−2.174; 1.966) | 0.921 | 1.802 (−0.307; 3.911) | 0.092 | 1.688 (−0.269; 3.644) | 0.089 |
| | Night-time birth | 1.105 (−0.540; 2.751) | 0.184 | 2.131 (0.495; 3.767) | **0.012** | 2.019 (0.488; 3.550) | **0.011** |
| | Gestational age (weeks) | −0.366 (−1.864; 1.131) | 0.626 | −0.139 (−1.726; 1.448) | 0.861 | –[c] | |
| | Infant's birthweight (kg) | −3.358 (−5.904; −0.812) | **0.011** | −2.828 (−5.422; −0.233) | **0.033** | −3.001 (−5.383; −0.618) | **0.014** |
| | Supplementary artificial milk feeds given | −0.612 (−2.359; 1.135) | 0.486 | −1.744 (−3.897; 0.409) | 0.110 | −1.857 (−3.454; −0.259) | **0.024** |
| | Exclusive breastfeeding at discharge | 0.964 (−0.687; 2.615) | 0.248 | 0.278 (−1.619; 2.174) | 0.770 | –[c] | |

Notes.
[a] First day: $R^2 = 0.334$; second day $R^2 = 0.390$.
[b] First day: $R^2 = 0.321$; second day $R^2 = 0.387$.
[c] Not in model.

in the first 24 h after birth and breastfeeding one to four months post birth. However, it must be acknowledged that differences in breastfeeding outcomes can also be explained by other differences in mothers' experiences in hospital, such as feeding policies (*Mattsson et al., 2015*). To reach firm conclusions regarding duration of SSC after birth and its associations with both short- and long-term breastfeeding, further studies are needed, in which data are collected about the total duration of SSC between mothers and infants after birth, and about all other relevant hospital practices related to the outcome variables.

Among fathers, mode of delivery was significantly associated with SSC during the first day, with longer SSC if the infant was born by caesarean section. Following caesarean birth, early SSC between mother and her newborn infant may be limited due to practical and medical safety reasons for both infant and woman (*Moore et al., 2016*). Data regarding infants' outcomes when cared for skin-to-skin with their father versus next to the father in a cot after caesarean birth have been evaluated (*Erlandsson et al., 2007*). The authors conclude that infants in the SSC group cried less and that their pre-feeding behaviour was

promoted compared with infants in the cot group. The latter may be attributed to our result showing that a mean duration of 4.4 h/day of SSC between infant-father the first 24 h following birth was associated with exclusive breastfeeding at discharge. Thus, it could be hypothesised that other caregivers than the mother can facilitate the development of the late preterm infant's early feeding cues.

More SSC during the second day for infants with lower birthweights was the only variable that was significant for both mothers and fathers. These results indicate that recommendations for international guidelines (*Nyqvist et al., 2010*) regarding SSC between low birthweight infants and their parents have, at least partially, been implemented in clinical practices in Sweden. In our study, the mean ± standard deviation total time spent with SSC with mothers the second day was 9.2 ± 7.1 h, whereas the corresponding figures for fathers were 3.1 ± 3.3 h. Although the reported time for SSC between infants and parents in this study is higher compared to reports from other settings in Scandinavia (*Olsson et al., 2012*), continuous SSC still remains an underused method of care for healthy late preterm infants in Sweden. This is underscored in our study as gestational age was not associated with infants' SSC with mothers or fathers during any of the first two days after birth.

It has been concluded that parents' satisfaction with support for SSC in postnatal care is associated with a larger extent of SSC during the first days after birth (*Calais et al., 2010*). Thus, nurses and midwives have a unique position to influence the use of SSC in clinical practice. However, nurses have expressed conflicting perceptions about SSC as they acknowledged its advantages for both infant and parents, but also questioned if the method is safe for the infants (*Blomqvist et al., 2013*; *Mörelius & Anderson, 2015*). In our study, no adverse event related to SSC was documented. Sudden unexpected postnatal collapse (SUPC) is a rare and dreaded complication that occurs among presumably healthy newborn infants within their first day of life. SUPC includes any condition resulting in temporary or permanent cessation of breathing or cardiorespiratory failure (*Pejovic & Herlenius, 2013*). Identified risk factors include prone position during SSC, unsupervised breastfeeding during the first two hours after birth, primiparity, and parents distracted by mobile phones (*Pejovic & Herlenius, 2013*). Consequently, specific policies and guidelines that address infants' safety during SSC are needed (*Chia, Sellick & Gan, 2006*; *Pejovic & Herlenius, 2013*). The implementation of such guidelines may in turn ease nurses' concerns regarding postnatal surveillance for late preterm infants receiving SSC and thus promote further use of the method. Inadequate staffing and education of staff on SSC has also been described as barriers to optimal implementation (*Koopman et al., 2016*) as well as visitors in the room during mothers' hospital stay (*Calais et al., 2010*; *Ferrarello & Hatfield, 2014*).

In the present study, SSC between the late preterm infant and the mother/father were measured and analysed separately. We can conclude that fathers provided SSC less than half the amount of time than mothers did and furthermore, fathers provided less SSC if supplemental methods of feeding were administered. Differences in gender roles may contribute to these findings, as mothers tend to take more responsibility for childcare and bond more strongly with a sick child compared to fathers (*Coffey, 2006*). Although

the infants in our study were defined as "healthy", it is possible that this explanation also applies to vulnerable newborns, i.e., late preterm infants.

Parity was associated with SSC with the father the second day after birth. Fathers provided 2.9 h longer SSC if the mother was primipara, compared to if the mother was multipara. It could be speculated that fathers were the primary caregiver to siblings during mothers' and late preterm infants' hospital stay. This is underscored by the fact that siblings were not allowed to stay overnight at the hospital. The 20th century has been characterised by social trends that have fundamentally changed the social context in which children develop, namely women's increased labor force participation and subsequently increased involvement of fathers (*Cabrera et al., 2000*). In Sweden, these trends are further underscored by the fact that 25 percent of the fathers share their parental leave with the mother, which, according to Swedish legislation, can be divided equally between the parents (*Statistics Sweden, 2016*). Although Sweden differs from other western countries with regard to parental leave, it is reasonable to assume that fathers' contribution to infant care will continue to increase in western societies during the present century. As a consequence, gender perspective needs to be implemented in clinical practice and research in paediatric care.

Infant's gender was associated with longer SSC among fathers for the first 24 h after birth, with 1.9 h longer SSC if the infant was a boy. This result is hard to interpret as studies investigating differences with regard to SSC between the four dyads of mother-daughter, mother-son, father-daughter and father-son is scarce. However, *Velandia, Uvnäs-Moberg & Nissen (2012)* conclude that mothers used more touching behaviour towards their newborn infant than fathers. Furthermore, mothers touched girls less than boys and fathers directed less speech towards girls than boys. There are psychoanalytic (*Washburn, 1994*) and gender theories (*Thompson & Walker, 1989*) suggesting that relationships within the family, e.g., the content or styles of interactions, are organised on the basis of gender. Others, on the other hand, argue that gender is only one of many factors influencing parent-child relationships (*Volling & Belsky, 1991*; *Russell & Saebel, 1997*). Our result raises questions of whether gender alone is important enough to create distinct relationships from birth between fathers and sons. However, further studies are needed to confirm whether an infant's gender influences duration of SSC with mothers and fathers, respectively among late preterm infants.

Our study had some limitations. An important factor to consider is that the sample included healthy late preterm infants born between 34 5/7 and 36 6/7 weeks of completed gestation. Consequently, the generalizability to late preterm infants as a group is limited. The sample size was too small to detect the extent of use of SSC when mothers suffered relatively rare birth complications, e.g., vacuum extraction, large perinatal tears and postpartum bleeding >1,000 ml, etc. National or even international research co-operations are needed to explore how to implement SSC among late preterm infants in these situations. Furthermore, our study did not include data concerning other comorbidities that may affect breastfeeding, for example, pregnancy-induced hypertension and diabetes. Additionally,

we did not collect any data regarding ethnicity. The latter is an important factor as socio-cultural context and norms influence SSC uptake (*Chan et al., 2016*) as well as breastfeeding rates and practices (*Jones et al., 2015*).

The response rate in the present study was 61%. However, parents of 17 infants (21%) did not complete the data collection about SSC. It could be speculated that these parents found the documentation in the diary complex and time-consuming. Another aspect to consider is that the data regarding SSC were based only on self-reports and that data from birth to study initiation was reported by the parents retrospectively. Our choice of data collection in the present study, i.e., the diary, was based on a previous study concluding that parents are able to document variables related to SSC of their infant in a complete and reliable manner (*Blomqvist, Rubertsson & Nyqvist, 2011*). Finally, the explained variation in the model, as given by $R^2$, was quite high, especially for fathers, with values of 33.4% for day 1 and 39.0% for day 2. However, due to the small sample size used, the results need to be interpreted with some caution.

## CONCLUSIONS

The present study raises questions regarding dose-response effects between duration of SSC and breastfeeding. Exclusive breastfeeding at discharge was significantly associated with infants' duration of SSC with fathers during the first day after birth, whereas no associations were found with mothers. Future studies are warranted that investigate duration of SSC between late preterm infants and their parents separately and its associations with breastfeeding. Taken together, a gender perspective needs to be implemented in paediatric research as the inclusion of mothers alone or parents as a couple may hinder firm conclusions regarding variables of importance for the infant's care.

### Funding
The authors received no funding for this work.

### Competing Interests
The authors declare there are no competing interests.

### Author Contributions
- Kerstin H. Nyqvist conceived and designed the experiments, wrote the paper, reviewed drafts of the paper.
- Andreas Rosenblad analyzed the data, contributed reagents/materials/analysis tools, prepared figures and/or tables, reviewed drafts of the paper.
- Helena Volgsten conceived and designed the experiments, performed the experiments, contributed reagents/materials/analysis tools, reviewed drafts of the paper.
- Eva-Lotta Funkquist conceived and designed the experiments, performed the experiments, reviewed drafts of the paper.

- Elisabet Mattsson conceived and designed the experiments, performed the experiments, wrote the paper, prepared figures and/or tables, reviewed drafts of the paper.

## Human Ethics

The following information was supplied relating to ethical approvals (i.e., approving body and any reference numbers):

Ethical approval to conduct the study was obtained from the Regional Ethical Review Board in Uppsala, Sweden, approval number 2010/287.

## Data Availability

The raw data has been supplied as Data S1.

## Supplemental Information

Supplemental information for this article can be found online at http://dx.doi.org/10.7717/peerj.3949#supplemental-information.

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
