# Peer review of "Early skin-to-skin contact between healthy late preterm infants and their parents: an observational cohort study"

_PeerJ, doi:10.7717/peerj.3949_

## Round 0.1 · original submission · Major Revisions

Some further comments in addition to those provided by the reviewers:

1. You have not described the study's limitations. Please provide a new paragraph on study limitations in the Discussion.

2. You have provided a very large number of p-values, which inflate the likelihood of observing Type I errors. Please make it clear in the Methods that no adjustments were made for multiple comparisons. This should also be referred to in the limitations paragraph in the Discussion, where you should state that the results need to be interpreted with some caution as a result.

3. You have carried out multivariable regressions using a very large number of predictors for a rather small study population. I would question the validity of this approach. I would have thought that it would be best to first run a series of univariate analyses (e.g. correlations and ANOVAs), and from these results select the parameters to be included in the multivariable models (based on p<0.05, p<0.1, p<0.2, or whatever threshold deemed appropriate). Alternatively, you could carried out stepwise regression analyses, but some statisticians believe these are more prone to errors. Note also that you need to consider possible issues of collinearity.

4. I am confused about your definition of late preterm. You have initially defined it in line 47, but the information in Table 2 appears to be contradictory as you seem to include infants born at 37 weeks of gestation as well. Or are you talking about infants born 36 0/7 to 36 6/7? If so, the information in the table is confusing, and I would remove the two categories, and only provide gestational age as means and SD.

5. I personally dislike the overuse of acronyms. I suggest therefore removing LPI and GA.

6. Please provide the age of fathers as well in your table describing sample characteristics.

7. There is no description of participants' ethnic characteristics. Ethnicity is likely to be an important factor influencing skin-to-skin contact. Were all mothers and fathers Caucasians? If so, were they all born in Sweden? It would be important to provide this information. If you don't have these data, you will need to discuss this issue as an important limitation in the Discussion.

8. Were parents made aware of the aims of your study? If they were aware that your primary outcome was length of time in skin-to-skin contact, wouldn't this have biased the results? In other words, as parents were aware that this was being measured, they intentionally or unconsciously could have increased direct contact with their babies.

Reviewer 1 ·

Basic reporting

please see attached report.

Experimental design

Please see attached report.

Validity of the findings

please see attached report

Annotated reviews are not available for download in order to protect the identity of reviewers who chose to remain anonymous.

·

Basic reporting

pass. see comments below

Experimental design

pass see below

Validity of the findings

pass see below

Additional comments

Thank you for allowing me to review this very well written manuscript on a population that is often ignored but have significant morbidity. Some minor editing is needed.
I have questions and concerns related to the retrospective nature of data collection during the first 24 hours. Additional information is needed regarding the retrospective data – how was it obtained, from where and by whom. It appears that you are comparing two different data sets (first 24 hours – retrospectively collected) and (2nd 24 hours prospectively collected by a parent diary). Please clarify and discuss what the problem with this may be (needs to be in the limitation section) and also included in the discussion section.
The discussion also needs to include possible reasons why BW only a predictor for SKS care in mothers and fathers during the 2nd day as well as why primiparity may affect the amount of SSC the father participates in.
Please include a limitation section which should included lack of data concerning other comorbidities that may affect lactation (ie pregnancy induced hypertension, diabetes etc)
Clarify why you consider the sample size to be satisfactory (line 297). Was a power analysis performed?
Line 28: I suggest using a different word than advice (perhaps recommend?)
Line 74: add the word “during” after respectively.
Line 79: I suggest using a more traditional description of the study design
Line 82: it would be more appropriate to report the number of LPI infants born at this hospital
Line 84: Define assistant nurse and midwifery care
Line 88: who offered the individualized support?
Line 91: please clarify whether placing infants STS immediately after birth was part of the research protocol or a standard of care. How did you verify this occurred? What if there were no father or partner? Was data collected regarding how soon after delivery SSC began?
Line 103: I would change twins for multiples to include triplets ect
Line 128-130: this is confusing and needs clarification
Information regarding consent and approval by the ethical review board needs to be presented earlier in the methodology
Line 151: define threatening fetal asphyxia
Line 178 to 180 is confusing and needs to be rewritten for clarity
Line 244: a definition of SUPC is needed
Line 277: “facilitated” should be substituted for another word

---

## Round 0.2 · Minor Revisions

As noted by the two reviewers, this revised version is much improved compared to the initial submission. I also generally concur with the reviewers' latest comments.

While I agree that the Discussion is a little wordy, I don't really see an issue with its current length. As PeerJ is in an online journal, I think the authors have the opportunity to publish a more extensive Discussion.

However, the manuscript needs careful editorial revision to correct grammatical errors and typos. For example:
- there is inconsistency in the spelling of "birthweight" or "birth weight".
- in the Abstract (under results), it should read "infant's birthweight" and "mothers" and "fathers" in the same sentence should be in singular form.
- in the Abstract and elsewhere in the text "dose response effect" (or similar cases) should read "dose-response effect".
- line 73 – it should say "breastfeed" instead of "breast feed"
- line 102 – delete the hyphen after "clinical"
- cases such as "36 6/7 weeks and days of gestation" or "36 6/7 gestational weeks and days" should read just "weeks of gestation".
- line 283 – best to say born "by" or "after" caesarean section, rather than "with".

In the References, the formatting of the journal names is inconsistent and should follow PeerJ's guidelines. I wonder if more information could be given regarding "Statistic Sweden (2016)" so that interested readers may be able to access it.

Figure 1: there is an issue with the flow of participants. There were 110 potential participants, but excluding the 18 who did not agree and 16 missed (total of 34 exclusions) should result in 76 participants instead of 86 as stated. Thus, I assume that the 16 missed should be only removed after the box with "agreed to participate" numbers, rather than beforehand.

Table 1:
- a space is missing before mothers in the heading.
- no need to repeat "gestational age" in the two rows below the subheading.
- the spelling of "breastmilk" should be consistent throughout the manuscript.

Tables 2 & 3:
- "associations" rather than "association"
- best to say "in the first and second"
- as per previous comment, if you refer to "infant's" SSC than "mothers" should be in its singular form.
- the spelling of "birth weight" or "birthweight" should be consistent throughout the manuscript.

Reviewer 1 ·

Basic reporting

Thank you for the opportunity to review the manuscript entitled: Early skin-to-skin contact between healthy late preterm infants and their parents: An observational cohort study.

I commend the authors for the improved clarity of the manuscript presented.

Abstract Background; the term breastfeeding you could quantify this for the reader as you report exclusive breastfeeding in the conclusion. I perhaps would add :
“its associations with breastfeeding (exclusive and any) ….

Results section Abstract; I suggest that the authors: use confidence intervals (CI) and not p values in the abstract. If using p values report with up to 2 significant digits and a maximum of 3 ( p=0.12, p=0.001) see: Lydersen, S. (2014). Statistical review: frequently given comments. Annals of the rheumatic diseases. 

Abbreviate the term standard deviation (SD).

Results section Abstract: it is quite difficult to make sense of so I would suggest that it is rewritten perhaps something like this;
Sixty-four mothers spent a mean 14.7 hours, 5.6, standard deviation (SD), during the first 24 hours after birth. While Fathers (n=64) experienced a mean of 9.2 hours 3.3 (SD) ….


Conclusion: line 53. Replace the word ‘it’ with SSC
Perhaps leave the reader with an important message about SSC and the timing of it and the importance of fathers and mothers both participating rather than restating the key results.
What should be done next?

Main text
Line 211- 214 – reference needed for the statistical methods used
line 215- please state if you adjusted for confounding factors/covariants and if so which ones and why?

Results
Tables: table 1 is not clear.
I suggest you combine table 1 (demographics) with 2 and 3 respectively. This will make it easier for the reader.
P values - report with up to 2 significant digits and a maximum of 3 (p=0.12, p=0.001)

Day
Variable
n (%)
Mean (SD)
Unadjusted
95% CI
Pvalue
Full model
95% CI
pvalue
Trimmed
95% CI
pvalue




Conclusions
A take home message is needed rather than just repeating the key results
What are the implications to practice / policy

Experimental design

no comment

Validity of the findings

Please review the conclusion; a take home statement is needed
what is the significance of the results -clinically and to policy?

·

Basic reporting

No comment

Experimental design

No comment

Validity of the findings

No comment

Additional comments

Thank you for allowing me to review the revised version of this manuscript. The vast majority of the reviewer’s comments were incorporated into the manuscript. There continues to be minor editorial changes needed. Other comments include the following;
Line 159-162 belongs in the introduction and not the methods section
Substitute the word children for infants.
The definitions of the demographic characteristics were defined previously in the methodology section and are redundant in the data analysis section.
Line 249: fetal asphyxia during delivery should be changed to fetal asphyxia. I am assuming that cardiotocography is fetal heart rate monitoring and should be stated as such. I also assume fetal PH is umbilical pH and should be changed to make the terms more easily understood.
Although use of a breast pump, cup, nipple shield were collected, they are not in the results section. Thus should be removed from the methods section.
No need to repeat content found in the table in the text

---

## Round 0.3 · Minor Revisions

Thank you for your latest revisions that have further improved the manuscript. However, both reviewers have noted some remaining minor revisions that should be addressed before the manuscript is formally accepted for publication.

Reviewer 1 ·

Basic reporting

generally clear however could be more succinct throughout.
See comments in the document provided
Needs to be edited carefully and consistent use of terms and abbreviations is needed.

Experimental design

see comments in table provided

Validity of the findings

low impact

Additional comments

See table provided for specific comments

Annotated reviews are not available for download in order to protect the identity of reviewers who chose to remain anonymous.

·

Basic reporting

No comment

Experimental design

no comment

Validity of the findings

no comment - these have been addressed in the limitation section

Additional comments

Thank you for allowing me to again review this manuscript on a very important topic regarding the late preterm infant who as we know is a high risk of morbidity. Although there are still minor editorial errors that need correcting the authors and adequately and appropriately responded to the reviewers comments and concerns. I just had a few comments/questions
1. Line 144: was phototherapy an exclusion criteria (would be rare but if the infant had a hemolytic process?)
2. Line 239: “minus” is confusing
3. Line 370: ethnicity is also important to rate of breastfeeding

---

## Round 0.4 · accepted · Accept

Please note that even though your article has been accepted for publication, there are some corrections that will need to be made before publication.

1. Make sure you carefully proofread your article as there are some issues that need fixing. For example, there is at least one instance where you say SCC instead of SSC and another of incorrect punctation (e.g. missing full stop in Abstract). Also, the expression of gestational age provided early in the methods need to be consistent with the rest of the manuscript.

2. There are issues with some references (e.g. Koopman et al, Raju et al). Thus, the reference list needs to be properly checked.

3. The file with the raw data is not really appropriate as it currently stands. Some changes that should be made include:
-All variables and the levels within them need to be translated to English.
-I assume #NULL refers to missing data, and such cells should probably be blank instead
-The levels for a number of parameters have > or <, which is surely wrong. One of the levels must be either ≥ or ≤
-You need to provide an associated data dictionary so that the parameters are easily identifiable and the levels within categorical variables correctly interpreted.